# Laser-Produced Cavitation Bubble Behavior in Newtonian and Non-Newtonian Liquid Inside a Rigid Cylinder: Numerical Study of Liquid Disc Microjet Impact Using OpenFOAM

**DOI:** 10.3390/mi14071416

**Published:** 2023-07-14

**Authors:** Amirhossein Hariri, Mohammad T. Shervani-Tabar, Rezayat Parvizi

**Affiliations:** 1Department of Mechanical Engineering, University of Tabriz, Tabriz 5166616471, Iran; hariri@tabrizu.ac.ir; 2Department of Cardiac Surgery, Shahid Madani Heart Hospital, Tabriz University of Medical Sciences, Tabriz 5163639889, Iran; rezayatp@gmail.com

**Keywords:** laser-produced cavitation bubble growth, laser-produced cavitation bubble collapse, a rigid cylinder, liquid disc microjet, OpenFOAM open source CFD software, water hammer impact, Non-Newtonian fluid

## Abstract

This study employs OpenFOAM to analyze the behavior of a single laser-produced cavitation bubble in a Newtonian/non-Newtonian fluid inside a rigid cylinder. This research aimed to numerically calculate the impact of liquid disc microjet resulting from the growth and collapse of the laser-produced bubble to the cylinder wall to take advantage of the cavitation phenomenon in various industrial and medical applications, such as modeling how to remove calcification lesions in coronary arteries. In addition, by introducing the main study cases in which a single bubble with different initial conditions is produced by a laser in the center/off-center of a cylinder with different orientations relative to the horizon, filled with a stationary or moving Newtonian/Non-Newtonian liquid, the general behavior of the bubble in the stages of growth and collapse and the formation of liquid disk microjet and its impact is examined. The study demonstrates that the presence of initial velocity in water affects the amount of microjet impact proportional to the direction of gravity. Moreover, the relationship between the laser energy and the initial conditions of the bubble and the disk microjet impact on the cylinder wall is expressed.

## 1. Introduction

The cavitation phenomenon has piqued the interest of researchers for over a century. This phenomenon has significant advantages besides its destructive effects, such as erosion, high noise, part damage, vibration, and efficiency loss, primarily occurring in engines and pumps. The dynamics of cavitation bubbles are commonly employed in various branches of engineering, including nanomaterials [1,2], chemical [3,4], mechanical [5,6], shipbuilding [7,8], ocean [9,10,11], environmental engineering [12], and important practical issues, such as surface cleaning [13]. In the medical field, this phenomenon can be used to break up kidney stones [14], deliver drugs and genes to cells [15], and treat cancerous tumors [16].

The first analytical studies on cavitation bubbles return to the dynamic studies of the spherical bubble, where the pressure field is always symmetrical around the sphere (like the conditions of the infinite environment). To this end, the Rayleigh model [17] investigated an empty bubble or a bubble filled with a gas in an inviscid and incompressible fluid and expressed the relationship between changes in the bubble’s radius and time as an ODE. Considering surface tension and fluid viscosity, the Rayleigh model was upgraded to the Rayleigh–Plesset model [18]. The effects of fluid compressibility were also formulated in the models of Gilmore [19] and Keller–Miksis [20]. In the study of the dynamics of non-spherical bubbles, the boundary integral method (BIM), finite element method (FEM), and finite volume method (FVM) have been used extensively.

Due to the potential for more precise flow analysis, FVM has been the basis for more recent studies. In the dynamics of non-spherical bubbles, the dynamic behavior of a single bubble or a group of bubbles is typically investigated near a boundary with varying geometries. Depending on the desired problem, the phenomenon of interest is analyzed from the growth stage of the cavitation bubble to its collapse, jet formation, and impact on the boundary. Free surfaces [21,22,23], solid surfaces [24,25,26,27], perforated plates [28,29], and elastic surfaces [30,31] are among the boundaries of interest for researchers. Various geometries are observed in the research, such as the investigation of the behavior of the jet formed by the cavitation bubble collapse in a rectangular channel [32], the study of the dynamics of a cavitation bubble in the middle of two parallel horizontal rigid walls with a vertically-closed rigid wall at one end [33], the investigation of the behavior of a laser-induced cavitation bubble near two rigid walls perpendicular to one another [34], and the dynamic analysis of a laser-induced cavitation bubble in the upper part of a rigid cylinder [35].

Cylinder geometry is discussed in this article as an important topic of interest to researchers. Sun et al. [36] examined the movement and evolution of cavitation bubbles in a cylinder during high-speed water entry (HSWE) using the fluid-solid interaction (FSI) method. Bao et al. [37] investigated experimentally the dynamics of a single cavitation bubble subjected to a transient acceleration within a water-filled tube. Rouzbahani et al. [38] evaluated the growth and collapse of a cavitation bubble inside a rigid cylinder with a compliant coating (a model of human vessels) using boundary integral equation methods and finite difference techniques. The elastic coating was represented as a membrane with a spring base.

Numerous studies have been conducted on the jet impact category caused by the cavitation bubble’s collapse into the wall. Zhang et al. [39] studied the influence of stand-off distance on the counter jet and high impact pressure on the dynamics of the laser-induced cavitation bubble near the wall. Tzanakis et al. [40] conducted an incubation pit analysis and calculated the hydrodynamic impact pressure caused by an acoustic cavitation bubble explosion near a solid boundary. Using the impact method, Rodriguez et al. [41] investigated the dynamic behavior of a collapsing bubble between two parallel and rigid walls. Ye et al. [42] first proposed a formula for the cavitation threshold and microjet speed to examine the effect of the microjet caused by the collapse of the acoustic bubble adjacent to the wall on the 1060 aluminum sheet. They found that cavitation occurs significantly in the liquid under the ultrasonic field because the applied ultrasonic pressure range is much larger than the liquid cavitation threshold.

In the current research, the growth and collapse of a cavitation bubble created by a laser inside a rigid cylinder are studied numerically to investigate the behavior of a microjet disc formed by bubble collapse, particularly the impact of a jet disc on the wall. It is demonstrated that, under particular initial conditions, the growth and collapse of the bubble, the formation of the disc liquid microjet, and the microjet impact proceed as depicted in Figure 1. Certain initial conditions indicate that if the initial pressure is low, the bubble may not reach the stage of collapse. After several growths and rebounds, it will either reach equilibrium or a disk microjet will form and disintegrate before reaching the wall.

OpenFOAM is used to perform the simulation, and modifications are made to the compressible InterFoam solver to enable the simulation of non-Newtonian fluid behaviors for the liquid phase, considering the effects of nonlinear compressibility in the liquid phase, as well as the use of the Noble–Abel equation of state for the gas phase. In Section 2, the theoretical model is presented. In Section 3, the specifics of the numerical model are discussed. The results and discussion are presented in Section 4, while the conclusion is presented in Section 5.

## 2. Theoretical Model

### 2.1. Bubble Model and Physical Assumptions

The current study assumes that the initial temperature of a laser-induced bubble is well below its boiling point. In addition, heat diffusion across the interface of the bubble is disregarded. Neglecting heat diffusion is justified by considering the experiments in Söhnholz [43]. Mass diffusion through the bubble wall can also be ignored, as the diffusion time scale is much greater than the time scale considered here for bubble dynamics. In addition, phase change effects are excluded because of their obscurity. The substance contained within the bubble is considered air, and the content of a bubble can be approximated as a non-condensable gas with constant mass undergoing adiabatic state changes. Surface tension, gas, and liquid viscosity are considered in this study. Their effect on the bubble dynamics in an asymmetric collapse adjacent to a solid wall affects the jet formation and dynamics [27]. Moreover, the gravitational effect is considered.

### 2.2. Characteristics of the Cavitation Bubble

In the current research, it is assumed that a laser in water creates a bubble. The laser-induced bubble can be considered relatively non-invasive compared to the spark-induced bubble, whose electrodes influence dynamics. In general, laser-induced bubble formation involves two distinct steps: plasma formation and plasma conversion to the gas/vapor content of the bubble. Because an electron can absorb many photons, the molecule may be ionized in the focal region of the laser in water with high photon densities. The free electron can then absorb bremsstrahlung when colliding with another molecule, inducing cascade ionization and plasma expansion [44]. The plasma formation and growth occur in the incredibly early moment of cavitation bubble generation, and this period of time needs a different equation of state describing a plasma. However, a bubble’s plasma-to-gas/vapor content conversion occurs in a matter of nanoseconds, and this brief period does not require special treatment [45]. 

After forming a cavitation bubble, its evolution can be precisely described using the relevant parameters and initial and boundary conditions. Due to the high initial pressure compared to the pressure of the surrounding environment and the existence of a pressure gradient in the radial direction, the initial single bubble, which is spherical at the beginning of its formation, grows, and its radius and volume increase with time. If the initial bubble is situated in an infinite environment, its growth is symmetrical and retains its spherical shape. As the volume of the bubble increases, the internal pressure gradually decreases and becomes less than the pressure of the surrounding fluid. The radial velocity of the bubble wall eventually reaches zero. The bubble then contracts due to reversing the pressure gradient’s direction. 

Multiple cycles of expansion and contraction cause the bubble’s radius to decrease at each stage due to viscous forces and energy loss. Eventually, the bubble reaches a radius of equilibrium. Due to the initial bubble’s proximity to the boundary, the bubble’s growth is asymmetric. The dynamics of the bubble will be entirely influenced by the geometry of the boundary and the proximity of the bubble’s center to the boundary due to the asymmetry of the bubble. Typically, the liquid disc microjet is directed toward the boundary in asymmetric conditions due to the pressure asymmetry surrounding the bubble.

### 2.3. Governing Equations

The VOF method [46] is effective for simulating the free surface flows of two immiscible liquids and a liquid and a gas. In this method, density *ρ*, viscosity *μ*, and thermal conductivity *k* for the entire field are expressed as functions of the physical properties of each phase and volume fraction α. The global density field $\rho \left(\vec{x},t\right)$ is defined as $\rho \left(\vec{x},t\right)=\alpha_{l}\left(\vec{x},t\right)\rho_{l}\left(\vec{x},t\right)+\alpha_{g}\left(\vec{x},t\right)\rho_{g}\left(\vec{x},t\right)$ where $\rho_{l}$ and $\rho_{g}$, are, respectively, the densities of the liquid and gas phases. The general viscosity field is also derived from the equation $\mu \left(\vec{x},t\right)=\alpha_{l}\left(\vec{x},t\right)\mu_{l}+\alpha_{g}\left(\vec{x},t\right)\mu_{g}$, where $\mu_{l}$ and $\mu_{g}$ are the dynamic viscosities of the liquid and gas phases, respectively. The global thermal conductivity field $k\left(\vec{x},t\right)$ is calculated from the equation $k\left(\vec{x},t\right)=\alpha_{l}\left(\vec{x},t\right)k_{l}+\alpha_{g}\left(\vec{x},t\right)k_{g}$ where $k_{l}$ the thermal conductivity of the liquid phase and the gas phase’s thermal conductivity.

In the above three equations, the interface’s position is implicitly determined by the transition $\alpha_{l}$ from 1 to 0. Using the VOF method, the fluid can be formulated separately with the density field $\rho \left(\vec{x},t\right)$, velocity field $\vec{U}\left(\vec{x},t\right)$, and pressure field $P\left(\vec{x},t\right)$, such that the continuity and Navier–Stokes equations are satisfied in the form of Equations (1) and (2):(1)$$\frac{\partial \rho }{\partial t}+\nabla \cdot \left(\rho \vec{U}\right)=0$$
(2)$$\frac{\partial \left(\rho \vec{U}\right)}{\partial t}+\nabla \cdot \left(\rho \vec{U}⊗\vec{U}\right)=-\nabla p+\nabla \cdot T+\int_{S\left(t\right)}\sigma \kappa \left({\vec{x}}^{\prime }\right)\hat{\vec{n}}\left({\vec{x}}^{\prime }\right)\delta \left(\vec{x}-{\vec{x}}^{\prime }\right)dS^{\prime }$$
where ∇ represents gradient, ∇⋅ denotes divergence, ⊗ is tensor multiplication, σ denotes the surface tension coefficient, κ is twice the average curvature of the interface of the $\hat{\vec{n}}$ vector perpendicular to the interface from gas to a liquid, $\delta \left(\vec{x}-{\vec{x}}^{\prime }\right)$ is the Dirac delta in three dimensions, ${\vec{x}}^{\prime }\in S\left(t\right)$ is a point on the interface, ${\vec{x}}^{\prime }$ is the point where the equation is evaluated, and *T* is the viscosity stress tensor of a Newtonian fluid defined as follows:(3)$$T:=\mu \left(\nabla \vec{U}+{\left(\nabla \vec{U}\right)}^{T}-\frac{2}{3}\left(\nabla \cdot \vec{U}\right)I\right)$$
where *I* is the unit tensor. Assuming there is no mass transfer between the gas phase (within the bubble) and the liquid phase (outside the bubble), the continuity Equation holds separately for both fluids:(4)$$\frac{\partial \left(\alpha_{i}\rho_{i}\right)}{\partial t}+\nabla \cdot \left(\alpha_{i}\rho_{i}\vec{U}\right)=0,\;\;i=l,g.$$

The energy equation is expressed as Equation (5):(5)$$\frac{\partial \left(\rho C_{p}T\right)}{\partial t}+\nabla \cdot \left(\rho \vec{U}C_{p}T\right)=\nabla \cdot \left(k\nabla T\right)+S_{T}$$
where $C_{p}$ denotes the specific heat at constant pressure, *T* is the temperature field, and $S_{T}$ represents the source term.

### 2.4. Equations of State

#### 2.4.1. Liquid Phase Equation of State

For the liquid phase and if the liquid surrounding the bubble is water, the Tait equation of state [47] for water is utilized—the Tait equation of state accounts for the nonlinear effects of compressibility.
(6)$$P\left(\rho \right)=\left(P_{\infty }+B\right){\left(\frac{\rho }{\rho_{\infty }}\right)}^{n_{T}}-B,$$

#### 2.4.2. Gas Phase Equation of State

Lofstedt et al. [48] demonstrated that the size of gas molecules affects severe bubble collapses, where the equivalent bubble radius reaches a few micrometers. To account for this effect, the co-volume is incorporated into the equation of state, and the resulting Equation, which is sometimes referred to as the Nobel bremsstrahlung Abel equation of state, is as follows:(7)$$R_{spec}T=P\left(\frac{1}{\rho }-\frac{\beta }{\rho_{n}}\right)$$

In the above Equation, *R_spec_* denotes the specific gas constant, *T* represents temperature, *β* is the co-volume, and $\rho_{n}$ is the equilibrium density of the bubble, which measures the gas mass inside the bubble. The following Equation can be derived assuming the adiabatic state change of the gas within the bubble:(8)$$P{\left(\frac{1}{\rho_{g}}-\frac{\beta }{\rho_{n}}\right)}^{\gamma_{g}}=const$$

For air, the ratio of specific heats is $\gamma_{g}=1.4$ and β=0.0015.

## 3. Numerical Model

### 3.1. Mesh Gridding

In the following sections, except for Section 4.4, a rigid cylinder with a diameter of 1 mm and a height of 20 mm filled with water is selected for the study. A laser bubble with initial radii of 0.15, 0.2, and 0.25 mm and pressures of 50, 65, and 80 MPa is assumed to exist inside the cylinder. This problem is modeled both as a complete cylinder, as depicted in Figure 2, and as a wedge with a 5° vertex angle, as shown in Figure 3. The complete cylinder model is used to compare with the wedge model and to investigate states in which axial symmetry is not established, such as when the cylinder is not upright and a bubble forms outside the cylinder’s center.

In the complete cylinder geometry, the local refinement region size is considered equal to the cylinder diameter, and using mesh trimming tools, including topoSetDict and extrudeMeshDict in OpenFOAM, the grid is refined to Δx=2.57
μm and Δz=2.85
μm. Outside the refined domain, the grid spacing increases with a progression factor of 1.12. The time step is set to 1 × 10^−14^ s, and the total number of cells is 6,545,760. Due to the substantial number of cells and the volume of calculations, OpenFOAM employs the parallel processing capability of eight processors.

A wedge with an angle of 5° is considered for states with axial symmetry. The initial geometry is divided into 40 radial segments and 400 vertical segments. Since the main phenomena occur in the center of the cylinder, the region above is locally trimmed using the OpenFOAM mesh trimming tools, including topoSetDict and extrudeMesh. The local refinement region size is considered 1.5 times the cylinder diameter, and the grid is refined to Δx=1.56
μm and Δz=1.63
μm. Outside the refined region, the grade spacing is Δx=12.5
μm and Δx=42.5
μm. The total number of cells in the wedge geometry is 389,680, and the time step is 1 × 10^−14^ s.

### 3.2. Initial Conditions

Due to the rapid nature of cavitation, it is difficult to determine the initial bubble conditions experimentally. Xie et al. [49] employed an approximate model to estimate the initial bubble conditions (temperature and pressure). The following Equation can express the bubble’s initial specific internal energy:(9)$$e=e_{0}+\Delta e$$
where $e_{0}$ denotes the initial specific internal energy of the liquid, e represents the initial specific internal energy of the gas, and Δe indicates the increase in the internal energy of the gas after absorbing the laser energy. The increase in internal energy is defined as follows:(10)$$\Delta e=\frac{E_{a}}{M}=\frac{E_{a}}{\frac{4}{3}\pi R_{0}{}^{3}\rho_{0}}$$
where $E_{a}$ denotes the laser energy the liquid absorbs and $\rho_{0}$ is the initial gas density inside the bubble. Similar to the research conducted by Zhang et al. [50], it is assumed that the laser energy is completely converted into bubble energy in this study. Assuming that the gas within the bubble is ideal, we have the following:(11)$$e=\frac{i}{2}vP=\frac{i}{2}RT$$

In the above Equation, *v* denotes the specific volume of gas, *i* is the degree of freedom of gas (i=6 for polyatomic molecules), *R* represents the specific ideal gas constant for water steam (R=461.5$\frac{J}{kg\cdot K}$), and $T_{0}$ is the initial temperature. Combining the equations above yields the following relations between bubble pressure and initial temperature:(12)$$P_{0}=\frac{2e}{iv}=\frac{2e_{0}}{iv}+\frac{3E_{a}}{2\pi R_{0}{}^{3}\rho_{0}iv}$$
(13)$$T_{0}=\frac{2e}{iR}=\frac{2e_{0}}{iR}+\frac{3E_{a}}{2\pi R_{0}{}^{3}\rho_{0}iR}$$

This study assumes that the laser pulse’s energy $E_{a}$ creates a focal region in the liquid with the radius $R_{laser}$, which becomes the initial bubble at the initial pressure $P_{init}$, radius $R_{init}$, and temperature $T_{init}$. Under these initial conditions, if the bubble is placed in an infinite environment, after several isentropic processes, it will reach the equilibrium radius $R_{n}$ and reference temperature of the environment. The laser energy is calculated in nine modes to produce a bubble with three initial radii of 0.15, 0.20, and 0.25 mm and at three pressure levels of 50, 65, and 80 MPa within a cylinder with a diameter of 1 mm and a height of 20 mm, respectively.

## 4. Results and Discussion

### 4.1. Experimental and Analytic Validation of Numerical Code

To validate the numerical code with laboratory results and analytical methods, bubble growth in an infinite environment is modeled and compared to Gilmore’s equation and experimental results. In Gilmore’s 1952 seminal paper [19], the author describes the development of the bubble radius in an infinite medium using the following ordinary differential Equation.
(14)$$\left(1-\frac{\dot{R}}{C}\right)R\ddot{R}+\frac{1}{2}\left(3-\frac{\dot{R}}{C}\right){\dot{R}}^{2}=\left(1+\frac{\dot{R}}{C}\right)H+\left(1-\frac{\dot{R}}{C}\right)\frac{R}{C}\frac{dH}{dt}$$
where $\dot{R}$ denotes the bubble wall velocity, *C* represents the instantaneous sound propagation velocity in the liquid in the bubble wall, and *H* is the enthalpy. Per the study cited above, this equation is accurate until $O\left({\dot{R}}^{2}/C^{2}\right)$.

According to the experimental findings of Han et al. [51], a bubble with an initial radius of 0.2 mm, an initial pressure of 10 MPa, and an initial temperature of 293 K will reach its maximum 1125
μm radius in a finite amount of time if placed in an infinite environment. Koch et al. [45] demonstrated that the field dimensions must be at least 100 times the maximum radius of the bubble to simulate an infinite environment. Simulating bubble growth in an infinite environment with the current code and considering the initial conditions of Han et al.’s research [51] leads to a bubble with a maximum radius 1115
μm in microsecond time 112.6
μs. Figure 4 depicts the dimensionless radius-dimensionless time diagram of the growth and first rebound of the bubble in an infinite environment, along with a comparison to the experimental results of Han et al. and Gilmore’s solution. Dimensionless radius is defined as R*=RRmax, and dimensionless time is defined as t*=ttRmax. It can be seen that there is a good agreement between numerical and laboratory results and the Gilmore model.

### 4.2. Simulating a Bubble’s Behavior inside a Rigid Cylinder While the Liquid Is Still

The growth, collapse, and dynamics of the disc liquid microjet were studied by creating three bubbles with initial radii of 0.15, 0.2, and 0.25 mm inside a cylinder filled with water with a diameter of 1 mm and a height of 20 mm at three pressure levels of 50, 65, and 80 MPa. The relationship between the initial values of radius and pressure is linear. The following Equation can be used to calculate the impact of the disk microjet on a non-rigid solid wall [52]:(15)$$P_{wh}=v\frac{\rho_{1}C_{1}\rho_{2}C_{2}}{\rho_{1}C_{1}+\rho_{2}C_{2}}$$
where *v* denotes the relative velocity between the microjet and the solid surface, $\rho_{1}$ and $C_{1}$ represent the density and velocity of sound in the fluid, respectively, and $\rho_{2}$ and $C_{2}$ are the density and velocity of sound in the solid, respectively. In the event that the solid wall is rigid, the $\rho_{1}C_{1}\ll \rho_{2}C_{2}$ relationship is established; thus, Equation (15) is simplified as follows.
(16)$$P_{wh}=v\rho_{1}C_{1}$$

For water, ρ1=998 kgm3 and C1=1483 ms [53]. Since the microjet velocity is equal to 0 on the rigid wall due to the no-slip condition, the velocity and, consequently, the microjet impact could not be calculated in the wall itself. Instead, the velocity and impact were calculated in three intervals of 96%, 98%, and 99% of the cylinder radius. Figure 5 shows the velocity and pressure contour for the growth and collapse stages of the bubble with a radius of 0.25 mm and a pressure of 80 MPa in seven stages. For the initial formation of the bubble, laser energy is calculated as. Under these initial conditions, the maximum radius 0.39 mm of the bubble is reached after 12.4 μs.

Equivalent radius refers to the radius of the sphere whose volume is equal to that of the deformed bubble. After reaching its maximum volume, the bubble begins to shrink and splits into two distinct areas at 17 μs. At this point, a liquid disc microjet with an initial velocity of 304.9 ms and a final velocity of 91.7 ms reaches the opposite bubble wall after 2.5 μs. At this stage, the bubble wall velocity is 10.4 ms. The microjet is located at 96, 98, and 99% distances from the cylinder’s center at 21.9, 22.3, and, respectively, and the water hammer impact generates 71.2 MPa, 69.9 MPa, and 52.7 MPa, respectively.

Due to the inclusion of the Tait equation of state for water, it is evident from the pressure contour that the pressure wave is also entirely captured. Figure 6 compares the impact of the liquid disc microjet on the opposite wall of the bubble and the impact at 96, 98, and 99% distances from the center of the cylinder. In each of the bubbles with initial radii of 0.15, 0.20, and 0.25 mm, the amount of water hammer impact on the opposite bubble wall and the distances of 96, 98, and 99% from the center of the cylinder increase almost linearly as the initial pressure inside the bubble increases from 50 to 80 MPa. In addition, for each bubble with an initial pressure between 50 and 80 MPa and an initial radius between 0.15 and 0.25 mm, the increase in the water hammer impact on the opposite bubble wall and the distances of 96, 98, and 99% from the center of the cylinder are approximately linear. It can also be observed that the percentage of change in impact with a change in radius is highest at 50 MPa pressure and lowest at 80 MPa pressure.

Table 1 summarizes the amount of energy and initial radius of the laser area that results in bubble formation, the initial conditions of the problem samples, and the final impact.

### 4.3. Simulating Bubble Behavior in a Rigid Cylinder Assuming a Flowing Liquid

In many real-world situations, we may be interested in creating a cavitation bubble inside a fluid-moving duct to take advantage of cavitation. Therefore, the growth and collapse of a cavitation bubble with an initial radius of 0.25 mm and an initial pressure of 80 MPa inside a cylinder with a diameter of 1 mm and a height of 20 mm is investigated at three velocities of 1, 2, and 3 mm/s and in two modes of upward and downward flows. Consequently, the difference in the water hammer impact of the disc microjets of the two mentioned states must be characterized by the state where the fluid is still. Figure 7 depicts the velocity and pressure contour for a bubble with an initial radius of 0.25 mm and an initial pressure of 80 MPa inside a vertical cylinder with a diameter of 1 mm and a height of 20 mm filled with water moving upward at a velocity of 3 mm/s. Figure 8 also illustrates the velocity and pressure contour for the growth and collapse phases of the bubble under previous conditions and a downward velocity of 3 mm/s in seven stages.

In Figure 9, the comparison of the hydrodynamic behavior of the cavitation bubble formed with an initial radius of 0.25 mm and an initial pressure of 80 MPa located in a vertical cylinder filled with water with a diameter of 1 mm and a height of 20 mm in three states of still water, water moving upwards with three velocities 1 mm/s, 2 mm/s and 3 mm/s and water moving downwards with three velocities of 1 mm/s, 2 mm/s, and 3 mm/s is conducted. A comparison of bubble behavior in still water and moving water in two states of bottom-up and top-down movement reveals that the growth and collapse processes of the bubble, the formation of liquid disc microjet followed by impact, for moving water (in both the bottom-up and top-down motion states), occur approximately 1–2 μs faster than the corresponding processes in still water, which occur at 22.5 μs. Due to the fluid’s initial velocity, the entire assembly’s kinetic energy is greater than that of a comparable system at rest, resulting in the bubble’s rapid growth and subsequent phenomena.

The presence of this energy causes the impact on the other bubble wall and the impact at distances of 96, 98, and 99% from the center of the cylinder to be greater in all six states where the fluid has an initial velocity than in the still fluid. However, the behavior of a fluid moving in the direction of gravity differs from its behavior when moving in the opposite direction. The numerical value of impact is the inverse of its magnitude. As the fluid velocity increases in a movement against the direction of gravity, the impact on the wall decreases at 96, 98, and 99% of the cylinder’s center. 

In contrast, as the velocity of a fluid moving in the direction of gravity increases, the amount of impact on the wall increases at distances above the cylinder’s center. It can be inferred that if there is movement in the fluid, gravity is significant, and the higher the flow velocity in the direction of gravity, the more intense the liquid disc microjet impact will be on the wall. Inversely, the greater the flow velocity against the direction of gravity, the weaker the impact of the wall.

In order to investigate cases in which axial symmetry is not established, the complete cylinder geometry was employed in this study rather than the wedge geometry. Figure 10 depicts the shape of the bubble and microjet in the perfect cylinder geometry, while Figure 11 compares the velocity contour for a bubble with an initial radius of 0.25 mm and an initial pressure of 80 MPa in the perfect cylinder geometry and the wedge geometry. The dimensions of the bubbles in two distinct geometries at different stages are identical. The wall impact time in the perfect cylindrical geometry is slightly longer than the corresponding time in the wedge geometry. In addition, despite the higher velocities in some small regions away from the boundary in the wedge geometry, in the bubble and microjet geometry, and the velocity at the time of wall impact, there is an exceptionally good match between the two geometries. Figure 12 compares the effects of the two geometries above. The impact size in the two geometries corresponds well.

When the bubble formed by the laser is not precisely in the cylinder’s center and has eccentricity, the entire cylinder geometry should be employed. Eccentricity is defined as the ratio of the distance between the bubble’s center and the cylinder’s center to the cylinder’s radius. Figure 13 depicts the shape of the bubble and microjet in a full cylindrical geometry assuming a 10% and 20% outlet from the center for the cavitation bubble with an initial radius of 0.25 mm and an initial pressure of 80 MPa. Figure 14 compares the velocity contour at the impact moments for Figure 13’s 10% and 20% eccentricity and the eccentricity-free condition. As is evident, the eccentricity mode has a shorter impact time than the eccentricity-free mode, and the microjet will be asymmetric. In addition, impact at 20% eccentricity occurs more quickly than impact at 10% eccentricity.

Figure 15 compares the impact caused by the collapse of a cavitation bubble in Figure 13 with 10% and 20% eccentricity and zero eccentricity. The amount of impact is greater under the condition of eccentricity compared to the condition without eccentricity. In eccentricity mode, the asymmetric disc jet has two branches; the branch closest to the wall has a higher velocity due to the shorter distance and makes a stronger impact on the wall within a shorter time. However, the behavior of liquid disc microjet at 10% and 20% eccentricity is distinct. Even though the bubble with a 20% eccentricity is closer to the wall, the disc microjet impact on the wall is less severe than that caused by a 10% center exit of the bubble. Due to the requirement of zero velocity at the boundary, the microjet velocity cannot increase beyond 10% eccentricity at 20% eccentricity; therefore, the velocity is lower, and its impact is less intense near the boundary. Therefore, it is possible to conclude that eccentricity increases impact intensity. However, the state of maximum impact will depend on the initial radius and pressure of the cavitation bubble.

Due to the gravity parameter’s effect on the lack of axial symmetry of the complete cylinder, this study also examines the effect of the installation angle of a cylinder on its vertical orientation. This is possible by analyzing the geometry of the complete cylinder. Figure 16 depicts the results of a dynamic simulation of a bubble with an initial radius of 0.25 mm and an initial pressure of 80 MPa inside a cylinder with a diameter of 1 mm and a height of 20 mm filled with still water at angles of 0°, 30°, 45°, and 60° concerning the horizon.

Figure 17 depicts the velocity contour in the microjet impact moment caused by the collapse of the bubbles in Figure 16 at 96, 98, and 99% of the cylinder’s center. By comparing the impact of the microjet at four angles of 0, 30, 45, and 60° to the vertical state in Figure 18, it is evident that changing the angle has no appreciable effect on the bubble dynamics and, consequently, the impact. In other words, it can be concluded that gravity does not affect the bubble dynamics in this geometry and under these initial conditions, assuming the water is still. This result is consistent with Blake et al. [54], reporting that gravity plays a minor role in bubbles with a radius of less than 1 cm.

### 4.4. Simulating the Behavior of the Bubble in Blood under the Assumption That the Vessel Wall within the Cylinder Is Rigid

Plaque is composed of fatty deposits that accumulate on the arterial walls. The arterial walls become rigid with plaque accumulation, and their internal path becomes more constricted. The obstruction can become more severe with sediment accumulation in areas such as the vein’s fork. This disrupts blood flow to other organs and tissues that require blood and nutrients. This phenomenon eventually causes atherosclerosis. This complication is initially accompanied by severe chest pain, also known as angina pectoris. If the feeding vessel is large and completely blocked, cell death (myocardial infarction) will occur, mitigating the severity of myocardial infection if treated promptly. 

Angiography is one of the conventional diagnostic methods for blood vessel occlusion complications; it determines the location and severity of the occlusion. In cases of limited obstruction, angioplasty is used to open the blockage, and a stent is placed to prevent atheroma plaque recoil. Typically made of titanium, a stent is an artificial mesh tube that, when placed in a vein, prevents the local constriction of blood flow [55]. This can result in an increase in plaque density in the vessel wall. Due to the impact of the disc microjet, one of the applications of cavitation bubbles is the separation of plaques attached to the vessel wall. Isolated plaques can be collected using a basket.

The numbers in Table 2 were used to simulate the expansion and collapse of a cavitation bubble and the formation of a disc microjet within the bloodstream [56,57]. The simulation used average systolic blood pressure of 120 mmHg and diastolic blood pressure of 80 mmHg to represent relative blood pressure.

Blood is a non-Newtonian fluid, and three models of Casson [58], Bird–Carreau [59], and Herschel–Bulkley [60] have been suggested to simulate it in various references. In the models cited, viscosity is calculated using Equations (17)–(19), respectively:(17)$$\nu ={\left(\sqrt{\frac{\tau_{0}}{\dot{\gamma }}}+\sqrt{m}\right)}^{2},\;\;\nu_{\min}\leq \nu \leq \nu_{\max}$$
(18)$$\nu =\nu_{\infty }+\left(\nu_{0}-\nu_{\infty }\right){\left[1+{\left(k\dot{\gamma }\right)}^{a}\right]}^{\frac{n-1}{a}}$$
(19)$$\nu =\min\left(\nu_{0},\frac{\tau_{0}}{\dot{\gamma }}+k{\dot{\gamma }}^{n-1}\right)$$

In Equation (17), $\tau_{0}$ denotes threshold stress, $\dot{\gamma }$ represents strain rate, *m* is the Flow Consistency Index, $\nu_{\min}$ and $\nu_{\max}$ are the minimum and maximum viscosity, whose values are suggested for blood as $m=3.935\times 10^{-6}\frac{m^{2}}{s}$, $\tau_{0}=2.903\times 10^{-6}\frac{m^{2}}{s^{2}}$, $\nu_{\min}=3.905\times 10^{-6}\frac{m^{2}}{s}$ and $\nu_{\max}=13.333\times 10^{-6}\frac{m^{2}}{s}$ [61]. In Equation (18), the values $\nu_{\infty }=1.32\times 10^{-5}\frac{m^{2}}{s}$, $\nu_{0}=3.3\times 10^{-6}\frac{m^{2}}{s}$, *k*=0.6046s, *n*=0.3742 and a=2 are considered for blood [59]. In Equation (19), the values $\tau_{0}=0.0175\frac{m^{2}}{s^{2}}$, $k=8.9721\times 10^{-3}\frac{m^{2}}{s}$ and n=0.8601 are suggested for blood [60].

In this study, to simulate the water hammer impact caused by the disc microjet impact formed by the collapse of the cavitation bubble inside the blood vessel, a vertical cylinder with a diameter of 6 mm (representing the diameter of the blood vessel) and a height of 120 mm is considered. Since the water hammer pressure when the disc microjet strikes the vessel wall should not be sufficient to cause tissue damage, it is necessary to determine the initial conditions of the bubble. Rigatelli [62] stated that pressures exceeding 3.9 MPa could cause vessel wall damage. To achieve this, the initial conditions of the cavitation bubble should be chosen so that the final impact at a distance of 99% of the cylinder’s radius is less than the acceptable value. In this simulation, a bubble with an initial radius of 1.5 mm was examined at initial pressures of 10 MPa and 12 MPa using the Casson, Bird–Carreau, and Herschel–Bulkley models. Figure 19 illustrates the velocity contour for the aforementioned issue at a pressure of 12 MPa. Figure 20 compares the impact of the disc microjet. As is evident, an excellent correlation exists between the outputs of the three methods.

## 5. Conclusions

This study investigated the behavior of a single cavitation bubble produced by a laser inside a rigid cylinder using the OpenFOAM software package. Specifically, the impact of a liquid disc microjet on the cylinder wall during the bubble’s collapse was analyzed. In the first step, a water-filled cylinder with a diameter of 1 mm and a height of 20 mm was chosen as a study sample for this purpose. The simulations were conducted in nine instances, including bubbles with three initial radii of 0.15 mm, 0.20 mm, and 0.25 mm and three initial pressure levels of 50 MPa., 65 MPa, and 80 MPa, precisely created in the cylinder’s center by the laser. Due to axial symmetry, the computational field of the wedge face with a 5° vertex angle was considered. 

In each of the bubbles with initial radii of 0.15 mm, 0.20 mm, and 0.25 mm, the amount of water hammer impact on the opposite wall of the bubble and at distances of 96, 98, and 99% from the center of the cylinder increases almost linearly as the initial pressure inside the bubble increases from 50 MPa to 80 MPa. It was also observed that the percentage of change in impact with radius variation is most significant at 50 MPa pressure and least at 80 MPa pressure. In the subsequent step, it was assumed that the water within the vertical cylinder is not static and moves in the direction of/opposite to gravity. A bubble with an initial radius of 0.25 mm and an initial pressure of 80 MPa was modeled inside a cylinder containing water with a diameter of 1 mm and a height of 20 mm, where the water flows with three velocities of 1 mm/s, 2 mm/s, and 3 mm/s in two upward and downward states. 

It was observed that bubble growth and collapse processes, which form liquid disc microjets followed by impact, are faster in moving water (in both bottom-up and top-down motion states) than in still water. In addition, the impact to the opposite wall of the bubble and at distances 96, 98, and 99% from the center of the cylinder are greater than the corresponding impact in the still fluid. It was also observed that in the movement against the direction of gravity, as the fluid velocity increases, the amount of impact on the wall decreases at distances of 96, 98, and 99% from the center of the cylinder. In contrast, in the movement in the direction of gravity, as the fluid velocity increases, the amount of impact on the wall increases at the distances above. Due to the axial asymmetric geometry, the computational field was modeled as a complete cylinder during the next step, which involved the simulation of the bubble dynamics created at points outside the cylinder’s center. 

It was revealed that the formation of a bubble outside the center of the cylinder results in the formation of an asymmetric liquid disc jet whose impact intensity is greater than the impact intensity of the liquid disc jet formed by the collapse of a cavitation bubble located in the center of the cylinder, and the impact process occurs in a shorter amount of time. In the subsequent step, the effect of the angle of the cylinder on its upright position was examined. The simulation was performed in the entire cylinder geometry for the dynamics of the bubble with an initial radius of 0.25 mm and an initial pressure of 80 MPa inside a rigid cylinder with a diameter of 1 mm and a height of 20 mm filled with still water at angles of 0, 30, 45, and 60° to the horizon, and it was observed that the amount of impact and the time of impact did not vary significantly with the angle. In still water, gravity is insignificant. 

Finally, to simulate the water hammer impact caused by the impact of a disk microjet formed by the collapse of a cavitation bubble inside a blood vessel, the corresponding simulation was performed inside a vertical cylinder with a diameter of 6 mm and a height of 120 mm, and according to the pressure level limitation of 3.9 MPa in vessel wall was observed to have a bubble with an initial radius of 1.5 mm and two pressure levels of 10 MPa and 12 MPa, providing water hammer impact within the allowable pressure range of the vessel wall. It was also observed that the three non-Newtonian fluid simulation models of blood, namely Casson, Bird–Carreau, and Herschel–Bulkley, have an excellent agreement.

## Figures and Tables

**Figure 1 micromachines-14-01416-f001:**
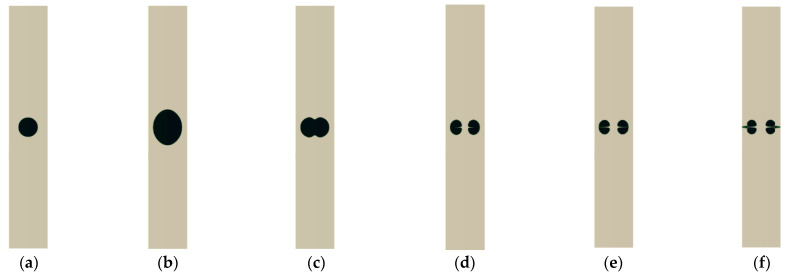
(**a**) Initial bubble formation, (**b**) bubble growth, (**c**) bubble contraction, (**d**) bubble collapse and disk microjet formation, (**e**) disk microjet impact on the other wall of the bubble, (**f**) disk microjet impact on the cylinder wall.

**Figure 2 micromachines-14-01416-f002:**
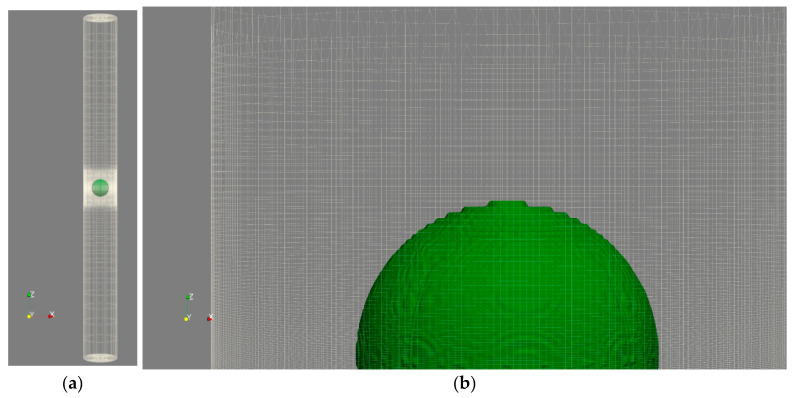
The solution field mesh in full cylinder mode, (**a**) the entire computational field, and (**b**) the reduced region adjacent to the bubble boundary.

**Figure 3 micromachines-14-01416-f003:**
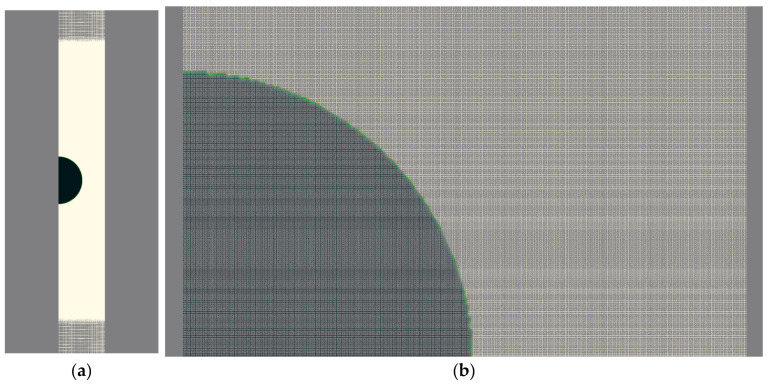
Mesh of the solution field in wedge mode, (**a**) a part of the computational field, and (**b**) the reduced region adjacent to the bubble boundary.

**Figure 4 micromachines-14-01416-f004:**
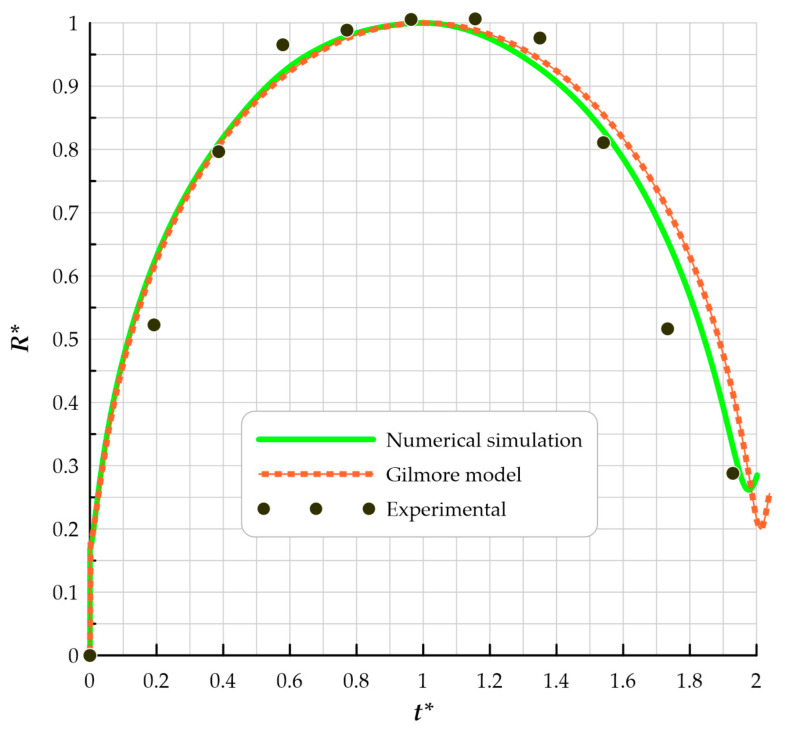
Comparison of dimensionless radius and time graphs for a bubble in an infinite medium.

**Figure 5 micromachines-14-01416-f005:**
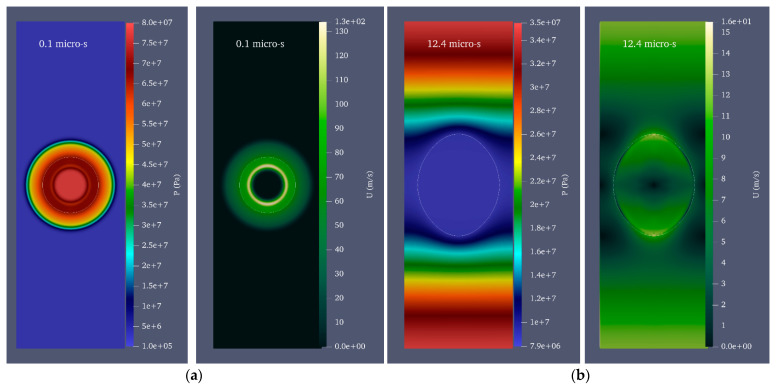
Velocity and pressure contour for a bubble with an initial radius of 0.25 mm and an initial pressure of 80 MPa located in a vertical cylinder with a diameter of 1 mm and a height of 20 mm filled with water; (**a**) the moment of the beginning of bubble growth, (**b**) the moment of maximum bubble growth, (**c**) the moment of liquid disc microjet formation, (**d**) the moment of disc microjet impact on the opposite wall of the bubble, (**e**) The moment of liquid disc microjet impact at 96% of the radius of the cylinder, (**f**) the moment of liquid disc microjet impact at 98% of the radius of the cylinder, and (**g**) the moment of liquid disc microjet impact at a distance of 99% of the cylinder radius.

**Figure 6 micromachines-14-01416-f006:**
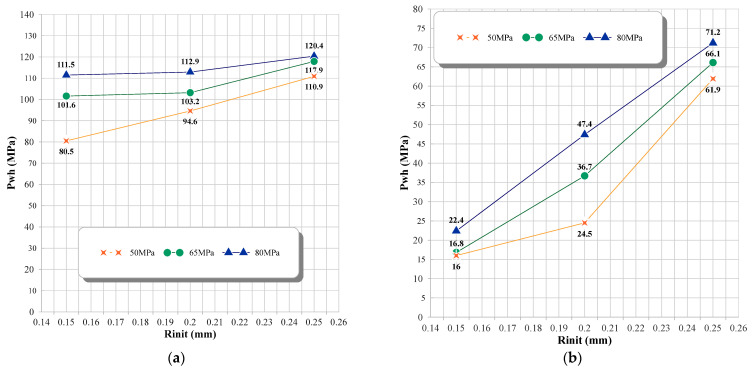
Water hammer liquid disc microjet impact in the bubble with three initial radii of 0.15, 0.2, and 0.25 mm and three pressure levels of 50, 65, and 80 MPa, inside a vertical cylinder with a diameter of 1 mm and a height of 20 mm, (**a**) compared to the opposite wall of the bubble, (**b**) at a distance of 96% from the center of the cylinder, (**c**) at a distance of 98% from the center of the cylinder, (**d**) at a distance of 99% from the center of the cylinder.

**Figure 7 micromachines-14-01416-f007:**
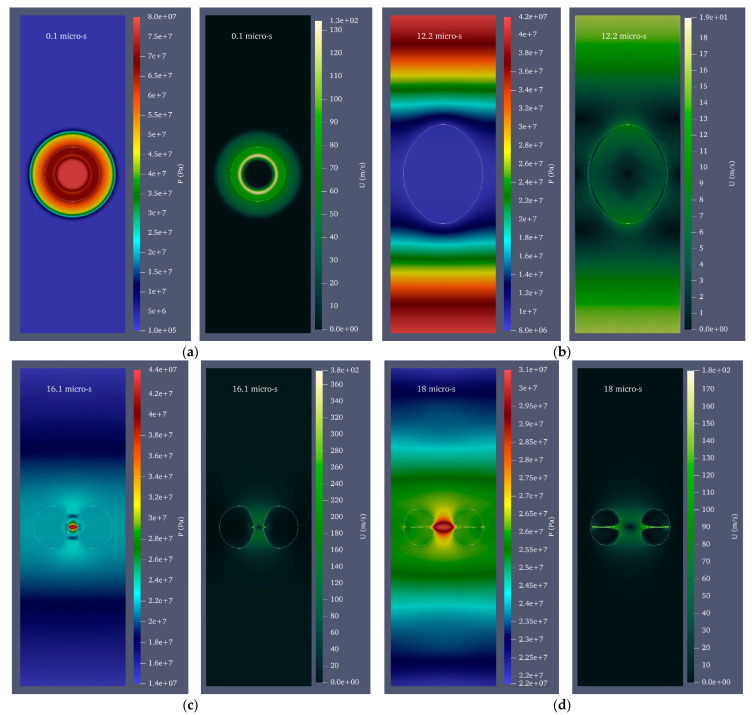
Velocity and pressure contour for a bubble with an initial radius of 0.25 mm and an initial pressure of 80 MPa in a vertical cylinder with a diameter of 1 mm and a height of 20 mm filled with water moving upward at 3 mm/s. (**a**) The beginning of bubble growth, (**b**) the moment of maximum bubble growth, (**c**) the instant liquid disc microjet formation occurs, (**d**) the impact of a disc microjet on the opposite bubble wall, (**e**) the moment of liquid disc microjet impact at a distance equal to 96% of the cylinder’s radius, (**f**) the moment of liquid disc microjet impact at a distance equal to 98% of the cylinder’s radius, and (**g**) the moment of liquid disc microjet impact at a distance equal to 99% of the cylinder’s radius.

**Figure 8 micromachines-14-01416-f008:**
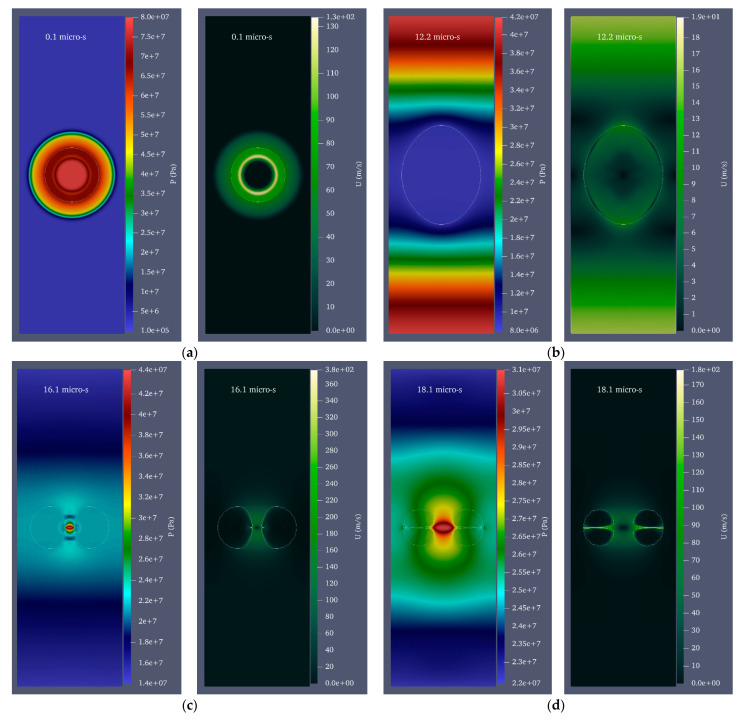
Velocity and pressure contour for a bubble with an initial radius of 0.25 mm and an initial pressure of 80 MPa in a vertical cylinder with a diameter of 1 mm and a height of 20 mm filled with water moving downward at 3 mm/s. (**a**) The moment when bubble growth begins, (**b**) the moment of maximum bubble growth, (**c**) the formation of a liquid disc microjet, (**d**) the liquid disc microjet's impact on the opposite bubble wall, (**e**) the moment of liquid disc microjet impact at a distance of 96% of the cylinder’s radius, (**f**) the moment of liquid disc microjet impact at 98% of the cylinder’s radius, and (**g**) the moment of liquid disc microjet impact at a distance of 99% of the cylinder’s radius.

**Figure 9 micromachines-14-01416-f009:**
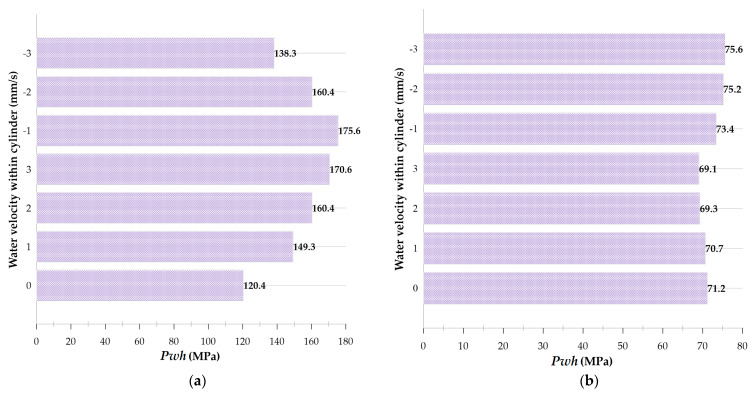
Comparison of liquid disc microjet impact caused by cavitation bubble collapse with an initial radius of 0.25 mm and an initial pressure of 80 MPa in a vertical cylinder with a diameter of 1 mm and a height of 20 mm filled with still and moving water at six different velocities in (**a**) the moment of disc microjet impact to the other bubble wall, (**b**) the moment of liquid disc microjet impact at 96% of the cylinder radius, (**c**) the moment of liquid disc microjet impact at a distance of 98% of the radius of the cylinder, and (**d**) the moment of liquid disc microjet impact at a distance of 99% of the radius of the cylinder.

**Figure 10 micromachines-14-01416-f010:**
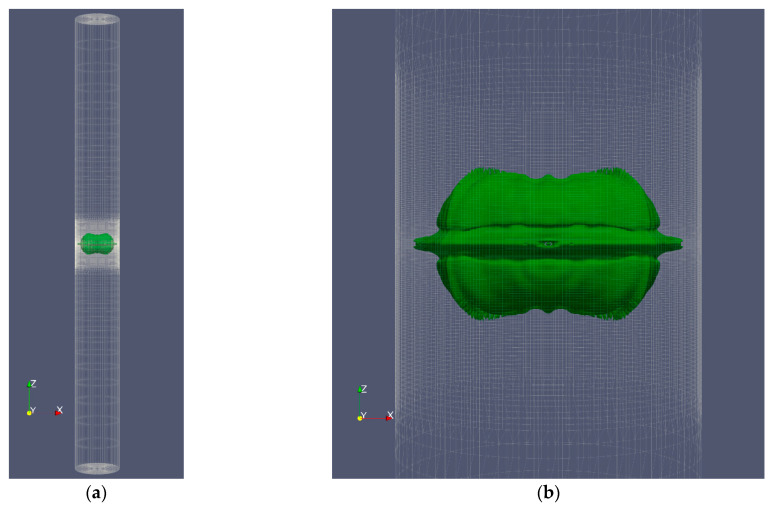
The shape of the deformed bubble and the microjet resulting from the growth and collapse of a cavitation bubble with an initial radius of 0.25 mm and an initial pressure of 80 MPa located in a vertical cylinder with a diameter of 1 mm and a height of 20 mm in a three-dimensional (3D) cylinder geometry, (**a**) the entire field, and (**b**) the area shredded around the bubble.

**Figure 11 micromachines-14-01416-f011:**
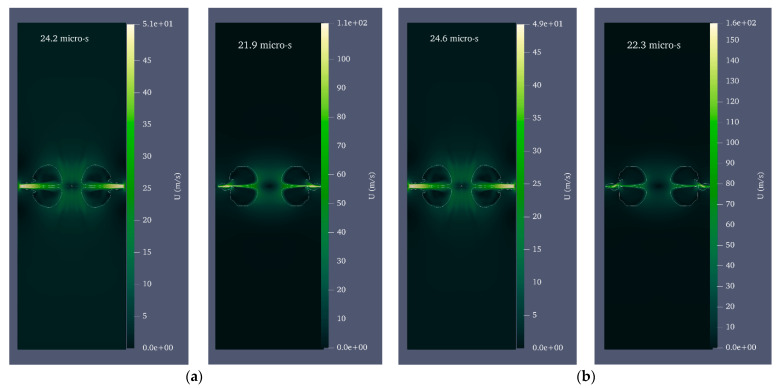
Comparison of the velocity contour for the bubble with an initial radius of 0.25 mm and an initial pressure of 80 MPa in full cylinder geometry (**left**) and wedge geometry (**right**) at the moment of impact, (**a**) 96% distance from the center of the cylinder, (**b**) 98% distance from the center of the cylinder, and (**c**) 99% distance from the center of the cylinder.

**Figure 12 micromachines-14-01416-f012:**
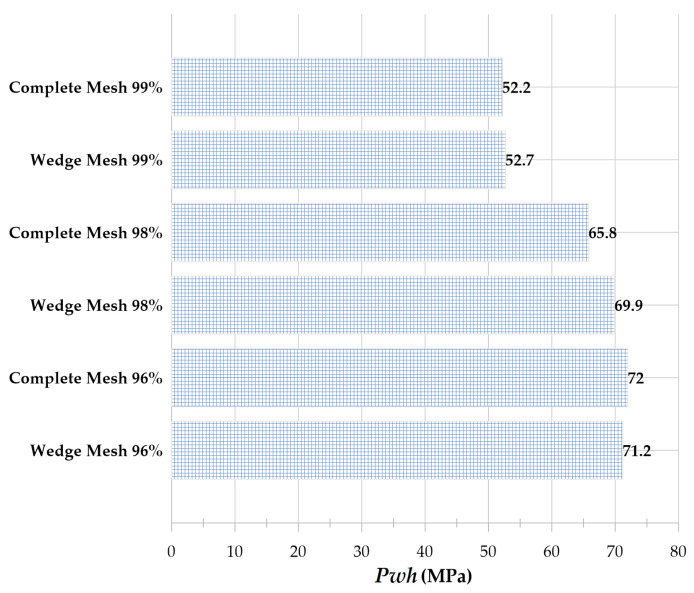
Comparison of liquid disc microjet impact at 96, 98, and 99% from the center of the cylinder for a bubble with an initial radius of 0.25 mm and initial pressure of 80 MPa in full cylinder geometry and wedge geometry at the moment of impact.

**Figure 13 micromachines-14-01416-f013:**
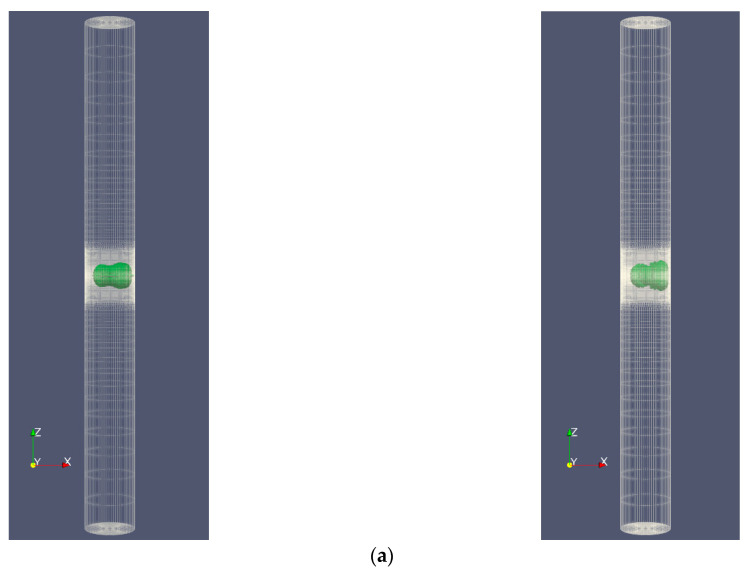
Deformed bubble shape and microjet in 3D cylinder geometry for a cavitation bubble with an initial radius of 0.25 mm and an initial pressure of 80 MPa located in a vertical cylinder with a diameter of 1 mm and a height of 20 mm with 10% eccentricity (**left**) and 20% eccentricity (**right**), (**a**) entire field, and (**b**) reduced area around the bubble.

**Figure 14 micromachines-14-01416-f014:**
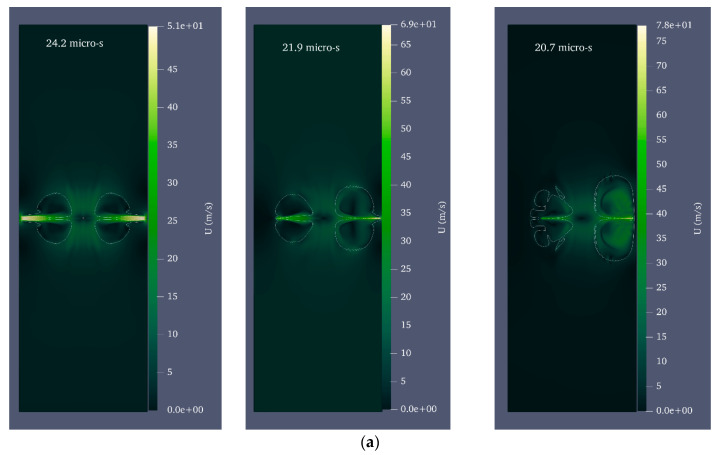
Comparison of velocity contour for the bubble with an initial radius of 0.25 mm and an initial pressure of 80 MPa without eccentricity (**left**), with 10% eccentricity (**middle**), and with 20% eccentricity (**right**) at the moment of impact, (**a**) 96% distance from the center of the cylinder, (**b**) 98% distance from the center of the cylinder, and (**c**) 99% distance from the center of the cylinder.

**Figure 15 micromachines-14-01416-f015:**
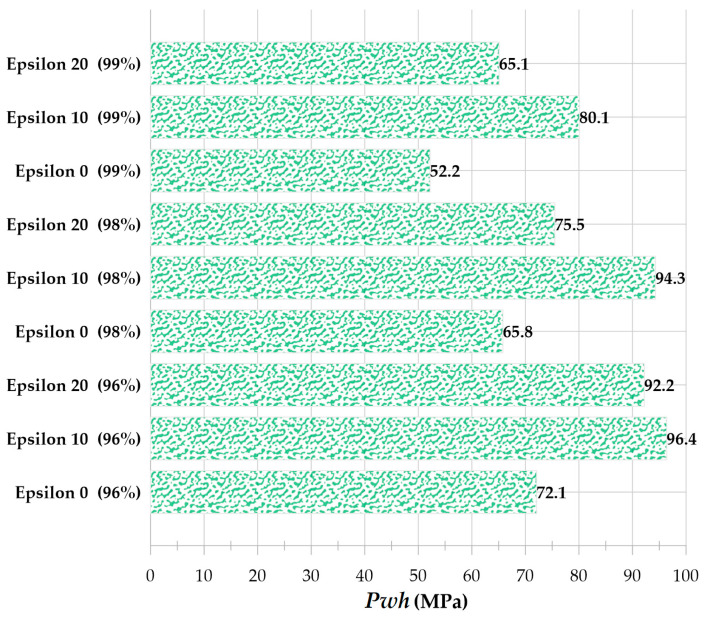
Comparison of liquid disc microjet impact for a bubble with an initial radius of 0.25 mm and an initial pressure of 80 MPa under three conditions: no eccentricity, 10% eccentricity, and 20% eccentricity at 96, 98, and 99% of the cylinder’s center.

**Figure 16 micromachines-14-01416-f016:**
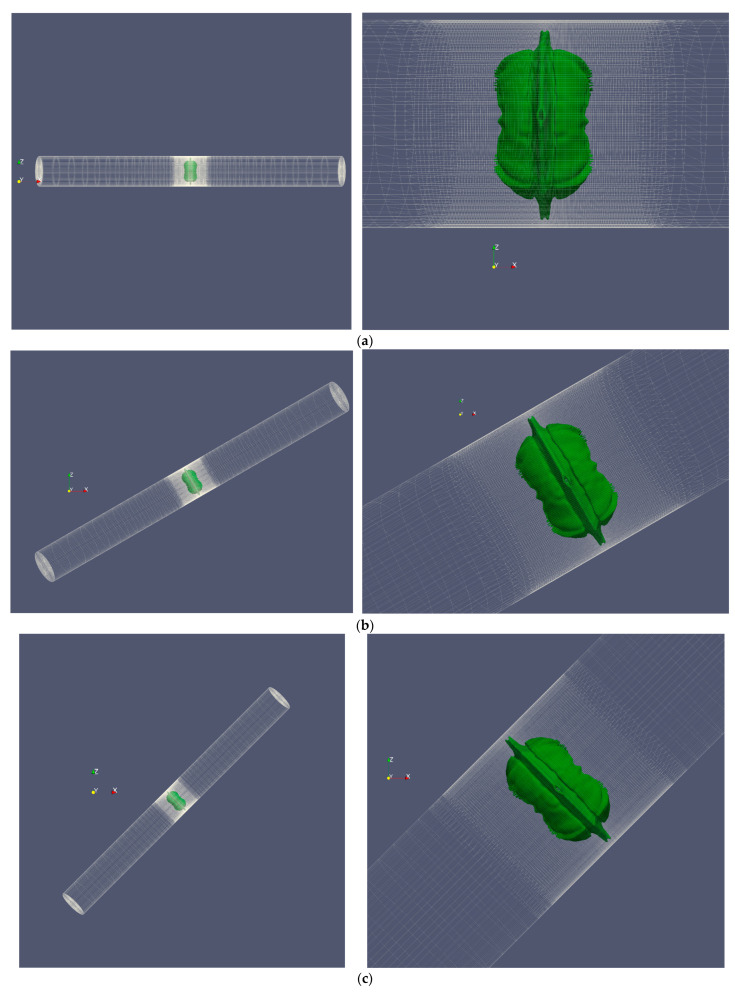
Deformed bubble and microjet caused by the collapse of the bubble with an initial radius of 0.25 mm and initial pressure of 80 MPa inside a cylinder with an initial diameter of 1 mm and initial height of 20 mm in the entire field (**left**) and the collapsed area around the microjet (**right**), (**a**) 0° angle to the horizon, (**b**) 30° angle to the horizon, (**c**) 45° angle to the horizon, and (**d**) 60° angle to the horizon.

**Figure 17 micromachines-14-01416-f017:**
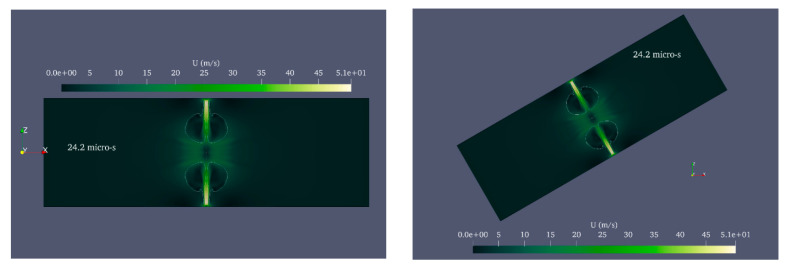
Impact of a microjet caused by the collapse of a bubble with an initial radius of 0.25 mm and an initial pressure of 80 MPa inside a cylinder with a diameter of 1 mm and a height of 20 mm at installation angles of 0, 30, 45, 60, and 90° relative to the horizon, (**a**) 96% distance from the center of the cylinder, (**b**) 98% distance from the center of the cylinder, and (**c**) 99% distance from the center of the cylinder.

**Figure 18 micromachines-14-01416-f018:**
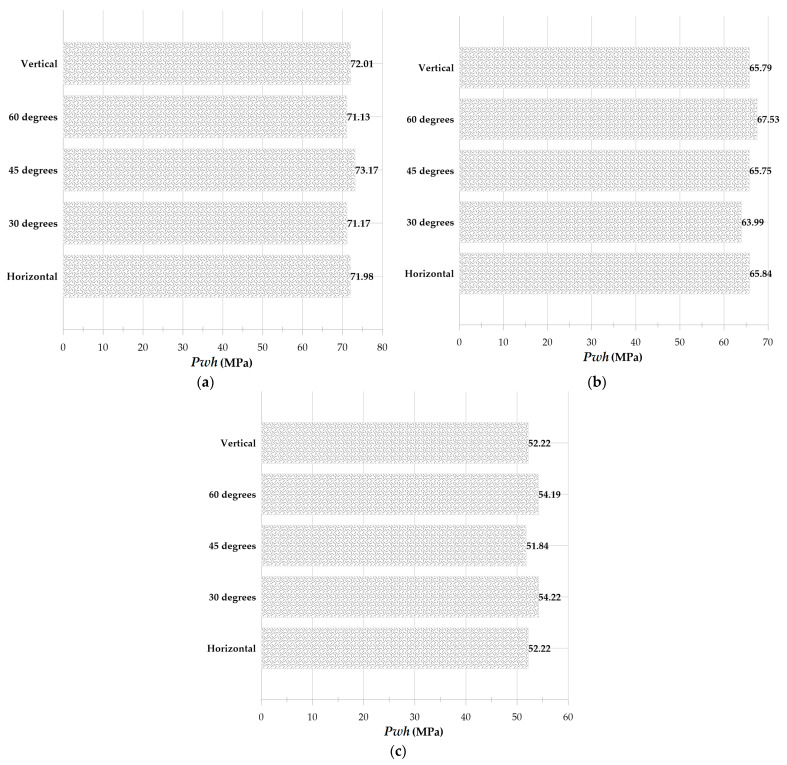
Comparison of disk microjet impact caused by the growth and collapse of a bubble with an initial radius of 0.25 mm and an initial pressure of 80 MPa inside a cylinder with a diameter of 1 mm and a height of 20 mm at angles of 0, 30, 45, 60 and 90° to the horizon, (**a**) 96% distance from the center of the cylinder, (**b**) 98% distance from the center of the cylinder, and (**c**) 99% distance from the center of the cylinder.

**Figure 19 micromachines-14-01416-f019:**
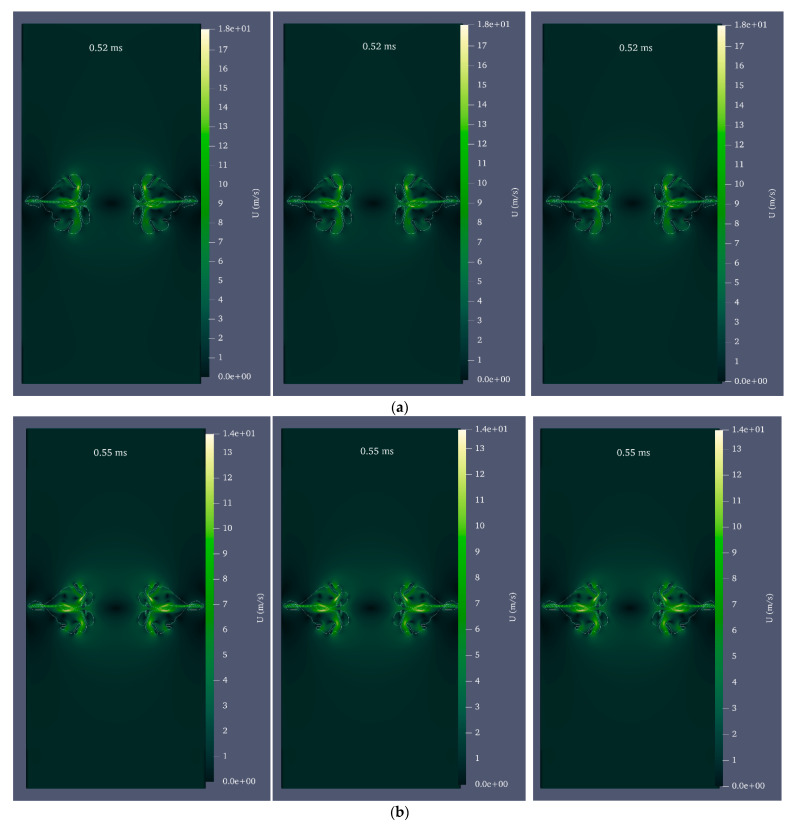
Velocity contour at the moment of liquid disc microjet impact for the growth and collapse of the bubble with an initial radius of 1.5 mm and an initial pressure of 12 MPa inside a vertical cylinder with a diameter of 6 mm and a height of 120 mm filled with blood using the Casson model (**left**), Bird–Carreau model (**middle**), and Herschel–Bulkley (**right**), (**a**) 96% of the cylinder’s center, (**b**) 98% of the cylinder’s center, (**c**) 99% of the cylinder’s center.

**Figure 20 micromachines-14-01416-f020:**
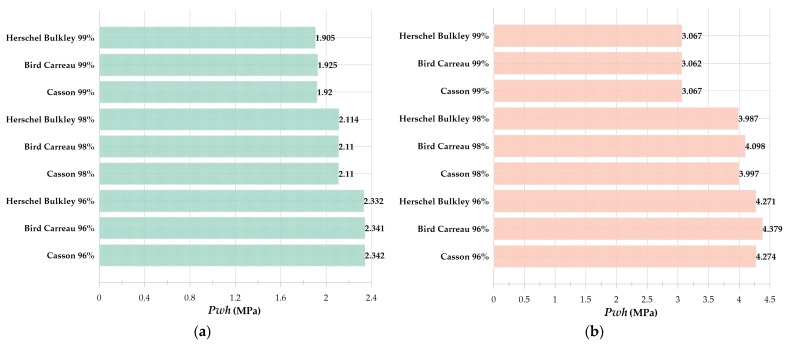
Comparison of microjet impact caused by bubble growth and collapse with an initial radius of 1.5 mm in three Casson, Bird–Carreau, and Herschel–Bulkley models at 96, 98, and 99% distances from a vertical cylinder with a diameter of 3 mm and a height of 120 mm, (**a**) initial pressure 10 MPa, and (**b**) initial pressure 12 MPa.

**Table 1 micromachines-14-01416-t001:** Initial conditions of laser, bubble, and water hammer impact at different distances.

*R*_*laser*_ (mm)	*E*_*laser*_ (mj)	*R*_*init*_ (mm)	*T*_*init*_ (K)	*P*_*init*_ (MPa)	Water Hammer Impact on Other Bubble Wall		Water Hammer Impact in 96% of Cylinder Radius		Water Hammer Impact in 98% of Cylinder Radius		Water Hammer Impact in 99% of Cylinder Radius	
					*Pwh* (MPa)	t (μs)	*Pwh* (MPa)	t (μs)	*Pwh* (MPa)	t (μs)	*Pwh* (MPa)	t (μs)
0.06	2.0	0.15	1721	50	80.5	23.1	16.0	45.1	9.3	50.4	3.7	52.7
0.06	2.6	0.15	1855	65	101.6	22.2	16.8	35.4	9.9	36.9	6.1	37.5
0.06	3.3	0.15	1969	80	111.5	21.6	22.4	32.7	17.3	33.5	14.3	34.3
0.08	4.8	0.2	1722	50	94.6	22.3	24.5	29.4	26.2	30.3	19.9	30.6
0.08	6.3	0.2	1856	65	103.2	21.2	36.7	27.8	30.4	33.4	21.8	36.3
0.08	7.7	0.2	1970	80	113.0	21.1	47.4	25.5	44.4	26.4	35.3	26.6
0.1	9.4	0.25	1722	50	111.0	21.5	62.0	25.1	52.6	25.5	46.9	25.8
0.1	12.2	0.25	1857	65	118.0	20.2	66.1	23.4	55.4	24.6	49.3	27.4
0.1	15.1	0.25	1970	80	120.4	19.5	71.2	21.9	69.9	22.3	52.7	22.5

**Table 2 micromachines-14-01416-t002:** Physical and thermophysical properties of blood.

pr	$\mu \;\left(cP\right)$	$C_{p}\;\left(\frac{J}{kg\cdot K}\right)$	$\rho_{\mathrm{blood}}\left(\frac{kg}{m^{3}}\right)$
21	$4.5\times 10^{-3}$	$3.21\times 10^{3}$	1050

## Data Availability

The datasets generated and supporting the findings of this article are obtainable from the corresponding author upon reasonable request. The authors attest that all data for this study are included in the paper. Data generated or the code used during the study are available from the corresponding author by request.

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
