# Peer review of "Laser-Produced Cavitation Bubble Behavior in Newtonian and Non-Newtonian Liquid Inside a Rigid Cylinder: Numerical Study of Liquid Disc Microjet Impact Using OpenFOAM"

_micromachines, 2023, doi:10.3390/mi14071416_

Round 1

Reviewer 1 Report

The physical assumption must be improved. especially the phase transition. 

Other characteristics of the cavitation bubble must be taken into account (conditions of creation and existing, collapse or expansion)

comparison to experiments would probably reveal different results and would show inaccurate assumptions and simplifications. I would consider that.

no comments on the English.

Reviewer 2 Report

The manuscript titled “Cavitation Bubble Behavior Produced by Laser in Newtonian 2 and Non-Newtonian Liquid inside Rigid Cylinder: Numerical 3 Study of Liquid Disc Microjet Impact Using OpenFOAM” by Hariri et al present a comprehensive study of cavitation bubbles inside rigid cylinders.

The manuscript does a good job establishing benchmarking comparisons with existing theory before moving into the presentation of new findings. It also does a good job justifying the use of wedges and symmetric considerations in the numerical computations by comparing them with simulations using the full geometry. The Dynamics are studied under different conditions including quiescent fluid and moving fluid both against and in the direction of the gravitational force and inclined cylinders. It also shows a case where symmetry is broken. Finally, the manuscript includes an application of the technique studying blood flow, where the blood is modeled as a non-Newtonian fluid using linear viscoelastic models.

In general, the modeling and numerical implementation presented in the manuscript are very strong and the figures and data provided are clear and complete.  However, the impact of the manuscript is greatly affected by grammatical errors and awkward wording (it made it very difficult to follow).  I suggest the manuscript undergoes extensive editing.

It really needs some editing, it is very good on the scientific part, but it fails in the description. 

Reconsider the paper after editing. 

Round 2

Reviewer 2 Report

The manuscript's writing was greatly improved.